# *Diaphorobacter nitroreducens* synergize with oxaliplatin to reduce tumor burden in mice with lung adenocarcinoma

Yalan Ni,[1,2] Rui Li,[1,2] Xiaoyu Shen,[1,3] Deli Yi,[1,3] Yilin Ren,[3] Fudong Wang,[3] Yan Geng,[4] Qingjun You[1,2]

**ABSTRACT** Lung adenocarcinoma (LADC) is the most common lung cancer and the leading cause of cancer-related deaths globally. Accumulating evidence suggests that the gut microbiota regulates the host response to chemotherapeutic drugs and can be targeted to reduce the toxicity of current chemotherapeutic agents. However, the effect of *Diaphorobacter nitroreducens* synergized with oxaliplatin on the gut microbiota and their impact on LADC have never been explored. This study aimed to evaluate the anti-cancer effects of *D. nitroreducens*, oxaliplatin, and their combined treatment on tumor growth in tumor-bearing mice. The composition of gut microbiota and the immune infiltration of tumors were evaluated by using 16S rRNA gene high-throughput sequencing and immunofluorescence, respectively. The inhibitory effect of the combination treatment with *D. nitroreducens* and oxaliplatin was significantly stronger than that of oxaliplatin alone in tumor-bearing mice. Furthermore, we observed that the combination treatment significantly increased the relative abundance of *Lactobacillus* and *Akkermansia* in the gut microbiota. Meanwhile, the combination treatment significantly increased the proportions of macrophage but decreased the proportion of regulatory T cells in the LADC tumor tissues of mice. These findings underscored the relationship between *D. nitroreducens* and the gut microbiota-immune cell-LADC axis, highlighting potential therapeutic avenues for LADC treatment.

**IMPORTANCE** Oxaliplatin is widely used as an effective chemotherapeutic agent in cancer treatment, but its side effects and response rate still need to be improved. Conventional probiotics potentially benefit cancer chemotherapy by regulating gut microbiota and tumor immune infiltration. This study was novel in reporting a more significant inhibitory effect of *Diaphorobacter nitroreducens* on lung adenocarcinoma (LADC) cells compared with common traditional probiotics and validating its potential as an adjuvant therapy for LADC chemotherapy in mice. This study investigated the impact of *D. nitroreducens* combined with oxaliplatin on the gut microbiota and immune infiltration of tumors as a potential mechanism to improve anticancer effects.

**KEYWORDS** *D. nitroreducens*, gut microbiota, immune infiltration, lung adenocarcinoma, 16S rRNA gene sequencing

Lung adenocarcinoma (LADC) accounts for approximately 40% of all types of lung cancer and carries the highest morbidity and mortality rate globally (1). LADC, as a type of non-small-cell lung cancer (NSCLC), typically occurs around the lung periphery and develops from mucosal glands. The cellular phenotype and the prognostic significance of the stromal component of lung tumors have been well studied. Although M2 macrophages are traditionally thought to contribute to tumor progression (2), recent studies have also shown that the infiltration of M2 macrophages is positively associated with favorable clinical outcomes in patients with NSCLC (3). Additionally, many studies reported that the high intra-tumoral infiltration of cytotoxic T cells (CD3$^+$ CD8$^+$) was

Address correspondence to Qingjun You, youqingjun@jiangnan.edu.cn, or Yan Geng, gengyan@jiangnan.edu.cn.

Yalan Ni and Rui Li contributed equally to this article. Author order was determined by their contribution.

The authors declare no conflict of interest.

See the funding table on p. 12.

associated with good prognosis and improved survival (4, 5). In contrast, the increase in the proportion of regulatory T (Treg) cells (fork-head box protein 3-positive, FOXP3$^+$) indicated a worse prognosis (6, 7).

Recent studies have recognized the gut microbiota as an essential regulator for innate and adaptive immunity, potentially modulating responses to chemotherapeutic agents such as oxaliplatin and immunotherapeutic agents such as anti-programmed death-1 antibodies (8–10). Some gut microbiota, such as *Bacteroides*, *Bifidobacterium*, *Akkermansia*, and *Faecalibacterium* spp., have garnered considerable attention for their beneficial effects on cancer therapy in preclinical tumor models and patients with cancer (9, 11–13). Moreover, many potential next-generation probiotics are currently developed using the latest-generation sequencing and bioinformatic platforms (14). These bacteria, including *Eubacterium limosum* (15, 16), *Enterococcus hirae* (8), *Enterococcus faecium* (17), *Collinsella aerofaciens* (18, 19), and *Burkholderia cepacian* (20), have demonstrated promising efficacy in promoting the anticancer effects. However, further exploration and evaluation are needed to elucidated the potential role of the microbiota in effectively modulating of anticancer treatment.

*D. nitroreducens* is an aerobic, Gram-negative bacterium belonging to the phylum Proteobacteria and family Comamonadaceae (21). It is ubiquitous in the environment and widely used in wastewater treatment due to its solid-phase denitrification capabilities (22). Despite its biotechnological applications, the bacterium was recently detected in clinical samples using high-throughput sequencing. For example, *Diaphorobacter* was detected in patients with pulmonary tuberculosis (23). *Diaphorobacter* was the characteristic intestinal microorganism in the neonates of pregnant women with gestational diabetes mellitus (24). It was also characterized as one of the bacterial microbiome components in the capsule of patients with capsular contracture (25).

Oxaliplatin is the third generation of platinum anticancer agents combined with 5-fluorouracil and leucovorin, which is routinely used to treat lung, ovarian, and colorectal tumors (26). However, peripheral sensory neuropathy of the extremities and gastrointestinal complications are common side effects of oxaliplatin (27). Therefore, as an effective chemotherapeutic agent, the response rate and side effects of oxaliplatin still need to be significantly improved. Currently, an increasing number of clinical trials using combination therapies are being conducted to find enhanced sensitization methods (28–30). With the development of high-throughput sequencing, the gut microbiota is increasingly recognized to play an essential role in determining chemotherapeutic efficacy and toxicity (31–33). The degree of pain hypersensitivity was significantly reduced in toll-like receptor 4 (TLR4) knockout mice induced by oxaliplatin compared with heterozygous littermates, confirming the effect of intestinal bacteria on chemotherapy toxicity (34). Compared with non-probiotic treatment, probiotic treatment with a mixture of probiotics, including *Bacillus mesentericus*, *Clostridium butyricum*, and *Enterocccus faecalis*, effectively alleviated oxaliplatin-induced intestinal and liver damage in mice (35). Despite studies examining the presence of *Diaphorobacter* in human lung and intestine tissues, how the gut microbiota is shaped and altered by *Diaphorobacter* and whether *Diaphorobacter* can synergize with oxaliplatin to improve the anti-LADC responses are still largely unknown.

This study aimed to explore the effects of *D. nitroreducens* synergized with oxaliplatin on LADC. We further characterized the composition of the gut microbiota and the phenotype of inflammatory infiltrate of tumor stroma.

## RESULTS

### *D. nitroreducens* inhibited the proliferation of A549 and H1975 cell lines

Recent studies revealed that the composition of gut microbiota was critical for the progression of cancer treatment (36, 37). First, the effects of five different microorganisms in our laboratory on the viability of LADC cell lines were examined. The data showed that four bacteria, *Lactobacillus paracasei* JN-1 (*L. par* JN-1), *Lactobacillus paracasei* E10 (*L. par* E10), *Lactobacillus rhamnosous* (*L. rha*), and *Escherichia coli* Nissle 1917 (*E. coli* Nissle

1917), did not have anti-tumor ability for the two LADC cell lines A549 and H1975 (Fig. 1A and B). However, *D. nitroreducens* (*D. nit*) significantly inhibited the growth of LADC cell lines, indicating that this bacterium had a good prognostic potential in LADC.

## *D. nitroreducens* synergized with oxaliplatin reduced tumor burden of LADC in mice

The oral supplementation of bacteria has attracted attention recently for its potential to modulate the anti-tumor response following immunotherapy (13). To validate whether *D. nitroreducens* modulated the response to LADC therapeutics, we first evaluated tumor growth and progression in mice-bearing A549 tumors treated with live *D. nitroreducens* alone or *D. nitroreducens* combined with oxaliplatin. From the 0th day after tumor inoculation, the mice were given *D. nitroreducens* via gavage, and starting from the 10th day, they received intraperitoneal injection of oxaliplatin (Fig. 2A). After combination treatment, the conventional A549-bearing mice exhibited an enhanced sensitivity to oxaliplatin, and tumor development was prevented (Fig. 2B). On day 40, the combination treatment group exhibited significant tumor growth suppression compared with the control or oxaliplatin treatment alone group. These results indicated that the combination treatment of *D. nitroreducens* with oxaliplatin had synergistic antitumor effects in A549-bearing mice. In addition, no significant differences were found in the changes in body weight among groups throughout the experiment (Fig. 2C). However, the spleen and liver weight exhibited a significant decrease in the combination treatment group compared with the control group (Fig. 2D and E).

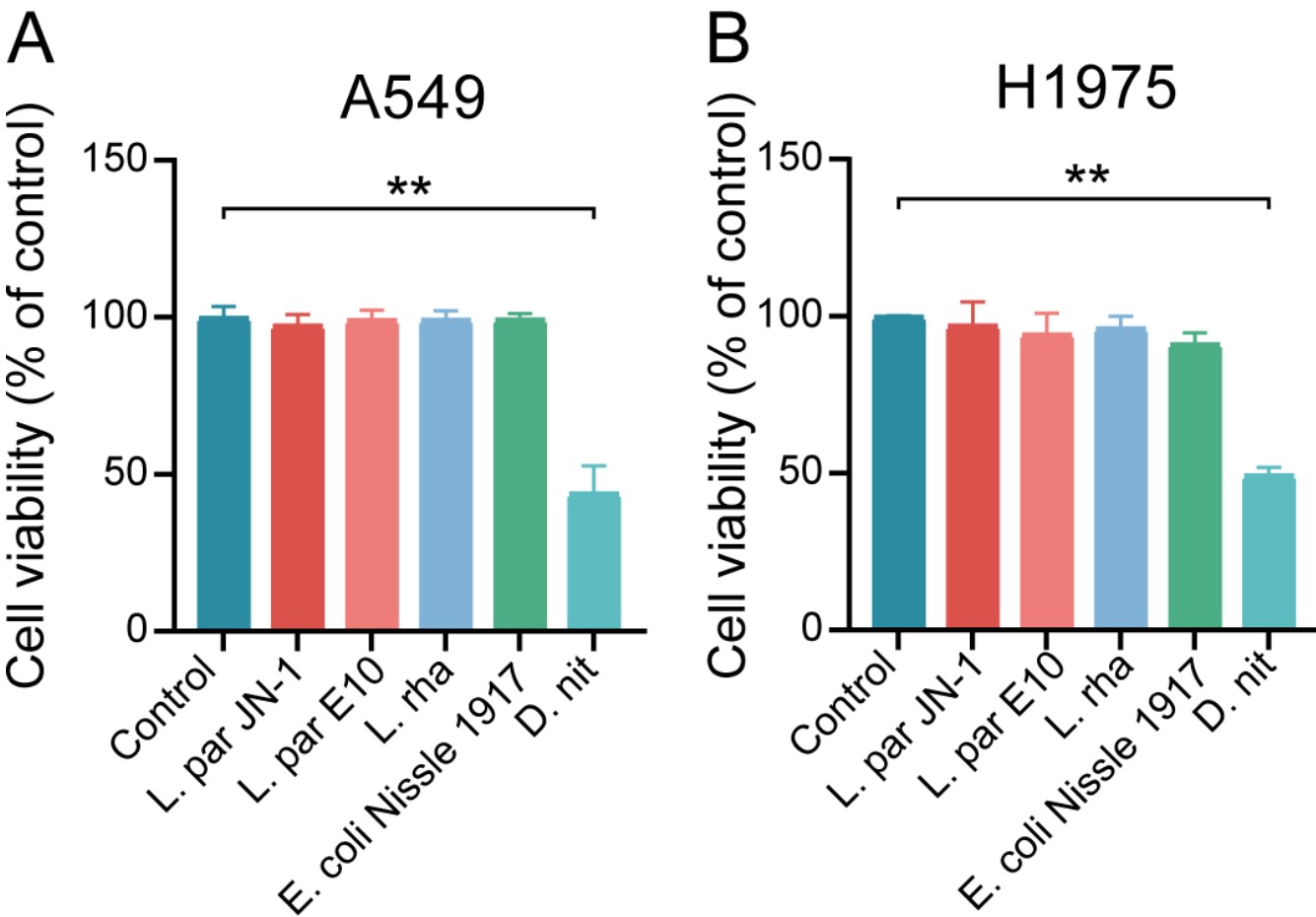

**FIG 1** *D. nitroreducens* inhibited the proliferation of A549 and H1975 cell lines. Effect of direct contact with multiplicity of infection of 100 of *L. par* JN-1, *L. par* E10, *L. rha*, *D. nit*, and *E.coli* Nissle 1917 on the survival rate of A549 (A) and H1975 cells (B).

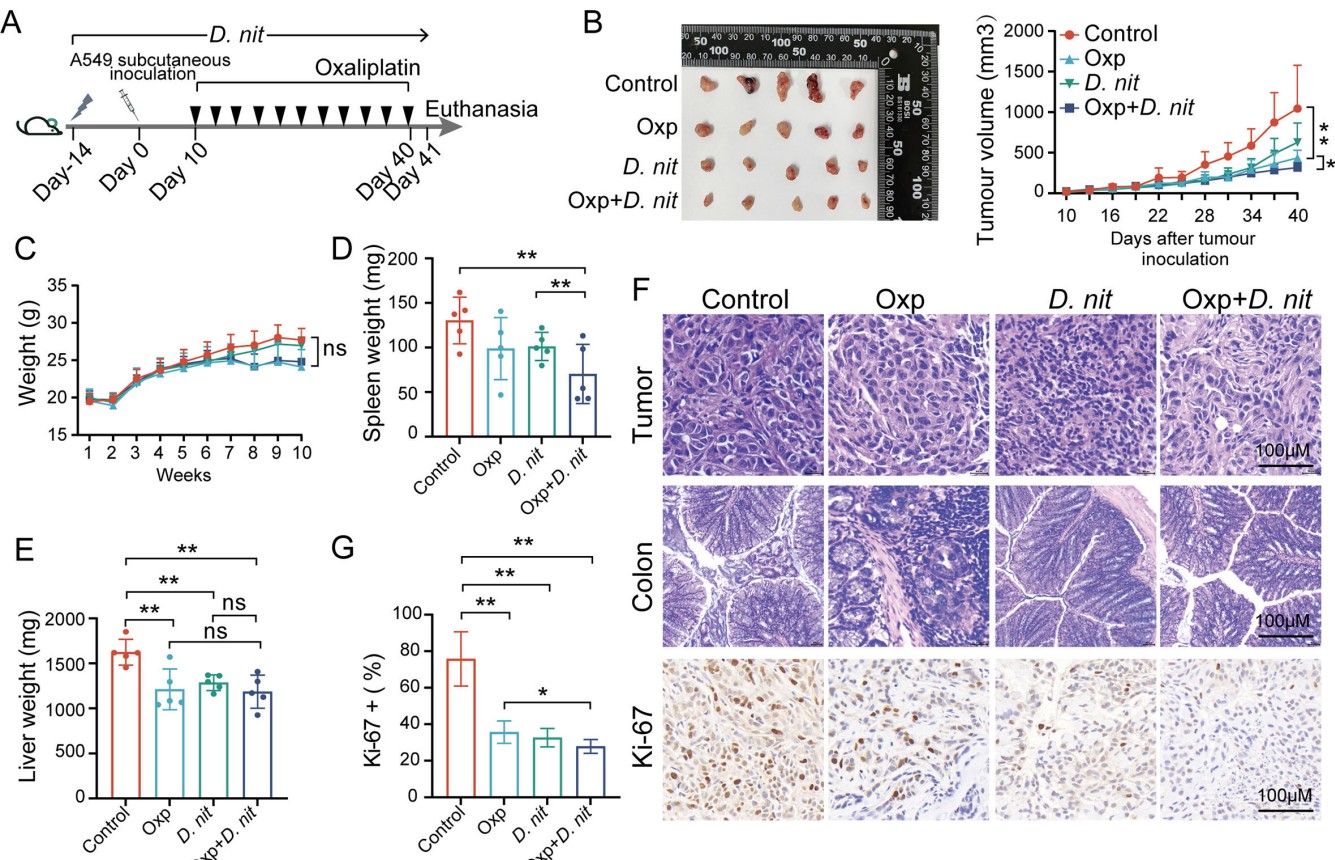

**FIG 2** Synergistic anti-tumor effect of the combination treatment of *D. nitroreducens* and oxaliplatin in the mouse tumor model. (A) Schematic of the experimental timeline: the mice were inoculated with A549 lung adenocarcinoma cancer cells 14 days after initial oral administration of *D. nitroreducens*, and oxaliplatin was intraperitoneally injected every 3 days. (B) Photos and tumor growth curves showing the changes in tumor volumes after administration of *D. nitroreducens*, with or without oxaliplatin. The number of mice per group was as follows: control, $n = 5$; Oxp (oxaliplatin treatment group), $n = 5$; *D. nit* (*D. nitroreducens* treatment group), $n = 5$; Oxp + *D. nit* (combination treatment with *D. nitroreducens* and oxaliplatin), $n = 5$. (C) Mouse weight growth curve. (D) Spleen weight. (E) Liver weight. (F) Hematoxylin and eosin staining on A549 subcutaneous tumors and colon (400×); immunohistochemistry staining of Ki-67 in A549 subcutaneous tumors of mice (400×). (G) Percent positive areas of Ki-67. All data are expressed as the mean ± standard deviation. *$P < 0.05$, **$P < 0.01$ (Student's *t*-test and one-way analysis of variance).

Furthermore, hematoxylin and eosin (H&E) staining revealed that the tumor cells in the control group were arranged neatly and tightly, with a large number of nuclei visible (Fig. 2F). After administration of *D. nitroreducens*, the tumor exhibited varying degrees of cancer cell damage, such as cell shrinkage and nuclear condensation. Meanwhile, compared with the control group, the combination treatment group exhibited more extensive damage, cell apoptosis, and scattered separated cells. In addition, H&E staining results also revealed that oxaliplatin caused colonic histological injury, but *D. nitroreducens* reversed the damage (Fig. 2F). However, all treatments displayed no effect on the histomorphology of the liver, as analyzed by H&E staining (Fig. S1). The results of Ki-67 staining showed that the combination treatment of *D. nitroreducens* and oxaliplatin significantly decreased the proliferation of LADC cells (Fig. 2G). The aforementioned results indicated that *D. nitroreducens* could augment the anti-tumor responses of oxaliplatin.

## Combination treatment of *D. nitroreducens* and oxaliplatin modified the gut microbial composition

Given that the gut microbiota has been reported to modulate the development of lung cancer and the efficacy of cancer immunotherapy (10), we investigated the effects of combination treatment of *D. nitroreducens* and oxaliplatin on the gut microbiota using 16S rRNA gene sequencing. First, we conducted alpha and beta diversity analyses to identify the differences in composition and diversity of gut microbiota among the four groups. No difference was found in the alpha diversity indicators, including Chao, Ace, Shannon, Simpson, and Sobs indexes, among the four groups (Fig. S2A). The beta diversity analysis indicated the extent of similarities and differences among microbial communities. Principal coordinate analysis (PCoA) of beta diversity based on an unweighted UniFrac distance revealed that both *D. nitroreducens* and oxaliplatin changed the gut microbiota structure in mice (Fig. 3A).

Then, we analyzed the taxonomic classification of microbiota using standard methods of assigning sequences to amplicon sequence variants (ASVs), which were clustered based on sequence similarity with existing rRNA databases. We next compared the enrichment of ASVs of four groups. The results showed a significant difference in the relative abundance of microbiota among different taxonomic levels. Four phyla (Firmicutes, Bacteroidota, Patescibacteria, and Desulfobacterota) were predominant in the four groups and accounted for up to 90% of the total observed sequences (Fig. 3B). Further comparison indicated that the oxaliplatin treatment group (Oxp group) had a higher relative abundance of Desulfobacterota but a lower relative abundance of Bacteroidota compared with the control group. Notably, *D. nitroreducens* significantly reduced changes in the abundance of Bacteroidota, Desulfobacterota, and Patescibacteria. Additionally, the relative abundance of Verrucomicrobiota was significantly higher in the combination treatment group compared with the other groups. At the family level, the combination treatment led to an increase in the abundance of Akkermansiaceae and Lactobacillaceae (Fig. S2B). At the genus level, treatment with *D. nitroreducens* combined with oxaliplatin significantly increased the relative abundance of *Lactobacillus* and decreased the relative abundance of *Clostridia_UCG-014* (Fig. 3C and D).

Next, we used linear discriminant analysis (LDA) and LDA effect size (LEfSe) algorithm to discover the differentially abundant biomarkers in the four groups (Fig. 3E). Based on LDA score, *Akkermansia* was the biomarker taxa in the combination treatment of *D. nitroreducens* and oxaliplatin. We confirmed that the relative abundance of *Akkermansia* significantly increased 10-fold due to the combination treatment of *D. nitroreducens* and oxaliplatin compared with that in the control group (Fig. 3F). These data indicated that the combination treatment of *D. nitroreducens* and oxaliplatin modulated the structure of gut microbiota and enriched the relative abundance of *Akkermansia*.

## Combination treatment with *D. nitroreducens* and oxaliplatin changed the gut microbial function

Altered gut microbiota composition might result in metabolic dysregulation or functional changes. Different treatments significantly changed the microbial community structure. Hence, we investigated the gut microbial functional pathways by phylogenetic investigation of communities by reconstructing unobserved states (PICRUSt2). As shown in Fig. 4, no significant difference was found between the control and Oxp groups. However, the microbial function was significantly changed by oral gavage with *D. nitroreducens*. Kyoto Encyclopedia of Genes and Genomes pathway analysis indicated that *D. nitroreducens* downregulated the methane metabolism, bacterial chemotaxis, nitrogen metabolism, biosynthesis of secondary metabolites, and microbial metabolism in diverse environments. Moreover, *D. nitroreducens* alone or in combination significantly downregulated fatty acid metabolism and biosynthesis (Fig. 4).

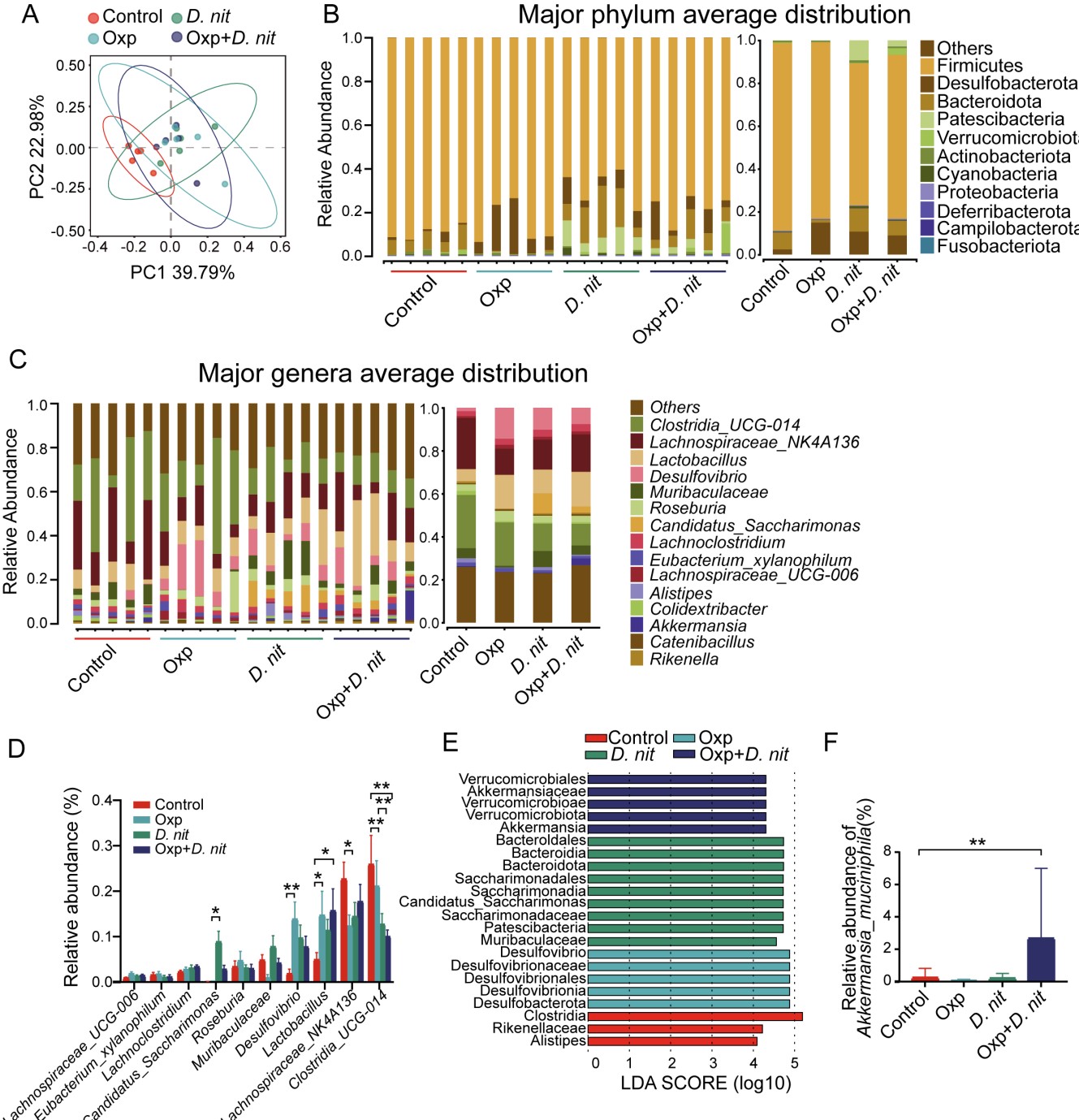

**FIG 3** Gut microbiota was changed by the combination treatment of *D. nitroreducens* and oxaliplatin in the mouse tumor model. (A) PCoA plot. (B) Bar plots showing the composition and relative abundance of each sample (left) and mean relative abundance in control, Oxp, *D. nit*, and Oxp + *D. nit* groups (right) at the phylum level. (C) Bar plots showing the composition and relative abundance of each sample (left) and mean relative abundance in control, Oxp, *D. nit*, and Oxp + *D. nit* groups (right) at the genus level. (D) Relative abundance of the top 10 genera in each group. (E) LEfSe analysis. Differences in the abundance of taxa among the control, Oxp, *D. nit*, and Oxp + *D. nit* groups were ranked according to their effect size by linear discriminant analysis. The taxa with an LDA score of more than 3.0 were identified as discriminative taxa among the groups. LDA score ≥4.0 and a *P*-value of <0.05 were considered to be significant. The degree of influence of species with significant differences among different groups was represented by the LDA score. (F) Relative abundance of *A. muciniphila* (%).

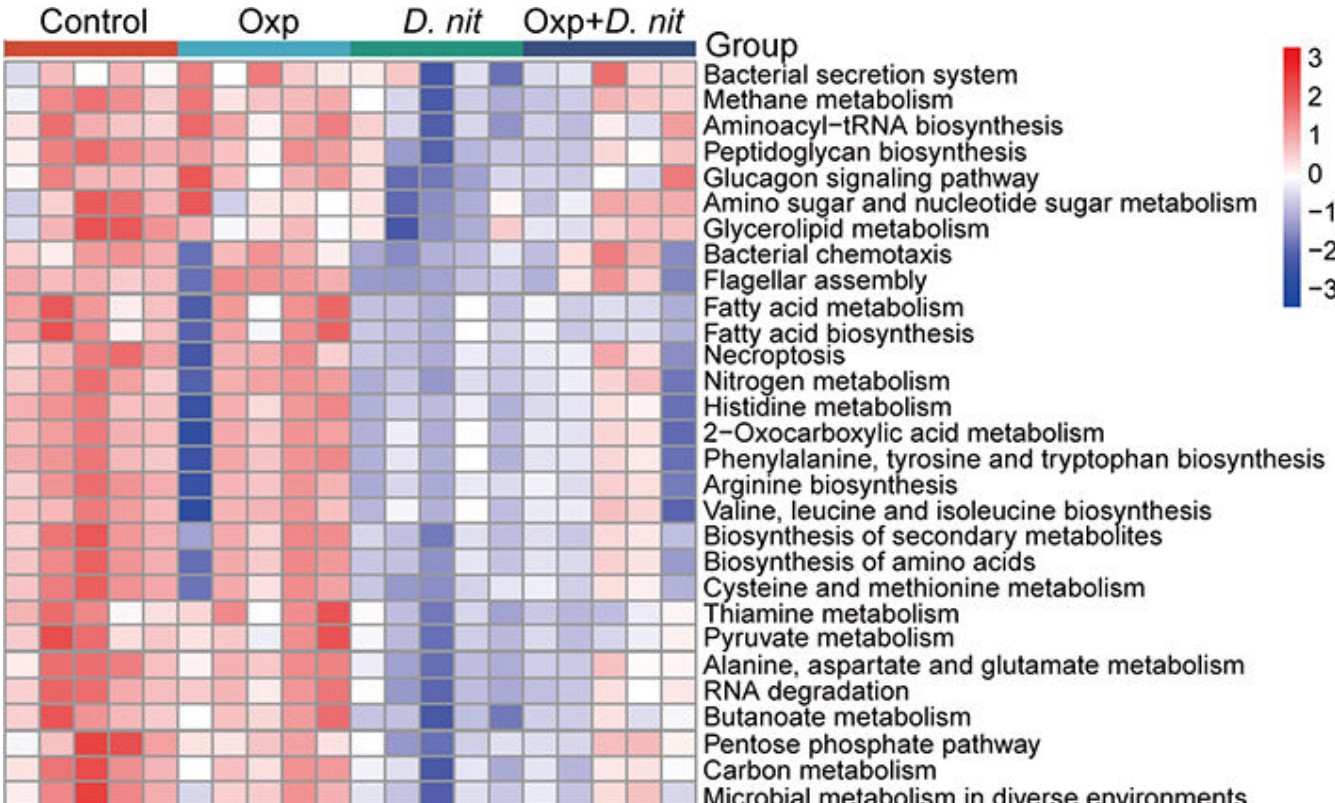

**FIG 4** The combination treatment of *D. nitroreducens* and oxaliplatin changed the prediction of metabolic function. Significant differences in metabolism pathway among the control, Oxp, *D. nit*, and Oxp + *D. nit* groups.

## Anti-tumor effects of combination treatment were associated with an increased number of macrophages and a decreased number of Treg cells

We next aimed to examine how the combination treatment affected immune cell responses and altered the tumor microenvironment. Tumor immune cell infiltration refers to the infiltration of immune cells from blood into the tumor tissue, which can be separated from the tumor tissue (38, 39). We found that the serum levels of tumor necrosis factor-α (TNFα) were significantly higher in mice treated with *D. nitroreducens* (Fig. 5A). Immune cells (macrophages and T cells) may represent the primary sources of pro-inflammatory TNFα (40). We furthermore double-stained macrophages (CD163[+] CD68[+]), helper T cell (CD3[+] CD4[+]), cytotoxic T cell (CD3[+] CD8[+]), and Treg cell (CD4[+] FOXP3[+]) of subcutaneous tumor tissues and examined the cells using immunofluorescence (IF) (Fig. 5; Fig. S3). Meanwhile, *D. nitroreducens* alone and the combination treatment significantly increased the number of CD163- and CD68-positive macrophages (Fig. 5B and D). However, the number of Treg cells (CD4[+] FOXP3[+]) significantly decreased in the tumors treated with *D. nitroreducens* and oxaliplatin (Fig. 5C and E). The number of other immune cells such as helper T cells (CD3[+] CD4[+]) and cytotoxic T cells (CD3[+] CD8[+]) was not influenced significantly in the tumor tissues (Fig. S3). *In vitro* experiments also demonstrated that *D. nitroreducens* promoted the secretion of TNFα in primary macrophages (Fig. 5F). These results suggested that *D. nitroreducens* and combination therapy led to fundamental remodeling of the tumor microenvironment through the action of macrophages and Treg cells.

## DISCUSSION

Recently, studies have provided increasing evidence of a close relationship between lung cancer and gut microbiota; probiotics have received much attention as adjunctive

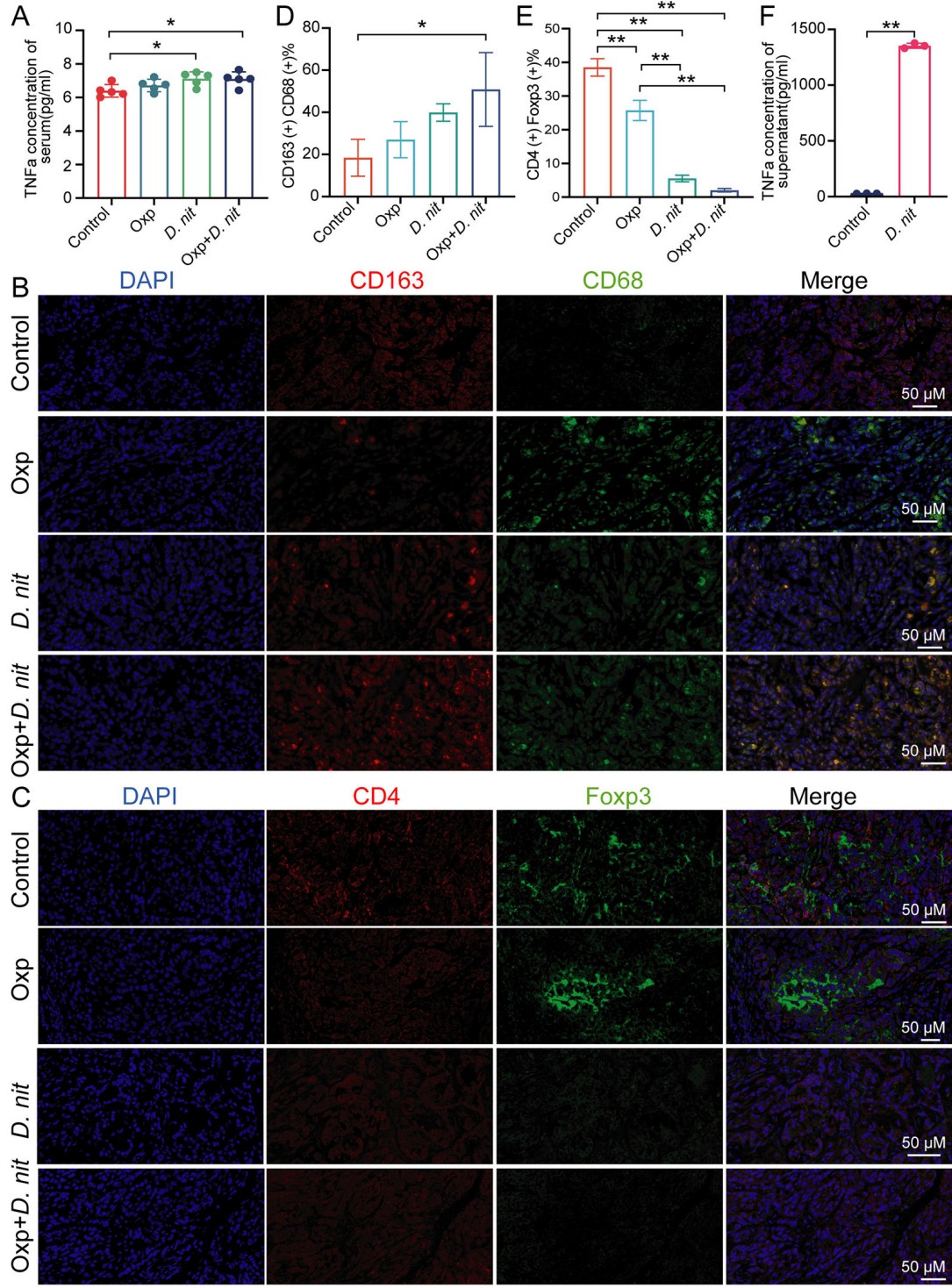

**FIG 5** Combination treatment of *D. nitroreducens* and oxaliplatin altered the numbers of macrophages and Treg cells in a subcutaneous tumor. (A) Concentration of TNFα in serum. (B) Immunofluorescence staining of the macrophage markers CD163 (pink) and CD68 (green) with 4,6-diamidino-2-phenylindole (DAPI) nuclear staining (blue), 40×. (C) Immunofluorescence staining of the Treg cells marker CD4 (pink) and Foxp3 (green) with DAPI nuclear staining (blue), 40×. (D) Quantification of double positivity of CD163 and CD68 cells. (F) Quantification of double positivity of CD4 and Foxp3 cells. (F) Concentration of TNFα in the supernatant of alveolar macrophages.

treatment of chemotherapeutic or immunotherapeutic agents (9, 10, 41). However, the relationship between *D. nitroreducens*, gut microbiota, and LADC has scarcely been

examined. In the present study, we observed that *D. nitroreducens* synergized with oxaliplatin to reduce the tumor burden of LADC in mice. Then, we found compositional differences in the gut microbiota between the oxaliplatin alone and combined treatment groups based on the 16S rRNA gene sequencing results. The overall structure of intestinal villi was restored in mice treated with probiotics (35); this phenomenon was also found in our study. Besides, the combination treatment of *D. nitroreducens* and oxaliplatin could increase the abundance of butyric acid-producing bacteria, such as *Akkermansia* and Ruminococcaceae. *Akkermansia* was associated with the differentiation of macrophages (42). The production of butyric acid and increased abundances of Ruminococcaceae were also observed in rats with cisplatin-induced nephrotoxicity prevented using other probiotics (43).

For functional analysis, the metabolism of fatty acid and the biosynthesis of amino acids were downregulated in the combined treatment group compared with the oxaliplatin alone group. The enhanced capacity to metabolize some amino acids of gut microbiota acted distally on the airway epithelium and protected against airway inflammation (44). The significance of fatty acid biosynthesis for cancer cell growth and survival has been confirmed by numerous studies (45, 46). Increased fatty acid biosynthesis could be a response to the high metabolic demand of cancer cells or an adaptation to reduced efficacy of serum-derived lipids in the tumor microenvironment (47, 48). Thus, the inhibition of fatty acid biosynthesis might be one of the anti-tumor mechanisms of *D. nitroreducens*.

Additionally, gut microbiota facilitates many chemotherapy-induced immune and inflammatory responses (32, 33). *Bifidobacterium* modified tumor-specific T cell induction and increased the number of T cells in the tumor microenvironment in patients treated with immunomodulators (9). A study also proposed the role of *Bifidobacterium infantis* in ameliorating oxaliplatin-induced toxicity by inhibiting helper T cell one and helper T cell 17 activation and enhancing cytotoxin regulatory T cell activity (CD4$^+$CD25$^+$Foxp3$^+$) in rats (41). *Lactobacillus* and segmented filamentous bacteria mediated the accumulation of helper T cell responses (49). *Diaphorobacter*, representing 79% of the Comamonadaceae family, was involved in T cell balance and then responded to allergens through direct links with TLR2 (50). Our results showed that *Diaphorobacter* promoted the infiltration of macrophages but reduced the number of Treg cells, ultimately slowing tumor progression. Many studies showed good prognostic significance for macrophages, whereas the accumulation of Treg cells in tumor suggested a worse prognosis (3, 7, 51, 52). The tumor microenvironment (TME) possesses various types of immunosuppressive cells including myeloid-derived suppressor cells, Treg cells, and tumor-associated macrophages (TAMs) (53). As one of the most abundant cell types present in TME (54), TAMs have been shown to exert important roles in the tumor therapy (55, 56). TAMs can be classified into the proinflammatory M1-type and anti-inflammatory M2-type macrophages (57). TAMs can exert a killing effect on tumor cells once activated by TNFα and interferon gamma (INF-γ) (58). Besides, there also is wide communication between TMAs and other immune compositions, such as Treg cells, cytotoxic T cells, neutrophils, dendritic cells, and so on (59, 60). This was consistent with our results, suggesting that the combination treatment of *D. nitroreducens* and oxaliplatin inhibited tumors by regulating the immune system.

However, our study had some limitations. We did not explore the specific ingredients of *D. nitroreducens* combined with oxaliplatin contributing to the inhibition of tumors in LADC. Also, this study did not fully explain how *D. nitroreducens* affected immune cells.

In summary, the combination of live *D. nitroreducens* and oxaliplatin reduced the tumor burden of LADC in mice with no obvious toxicity, modified the gut microbial composition, and promoted the infiltration of macrophages and Treg cells in tumor tissues. Our findings suggested that *D. nitroreducens* might be considered a relatively new and valuable adjunct agent to chemotherapy in LUAD.

## MATERIALS AND METHODS

### Bacteria

*D. nitroreducens* (CCTCC#AB2020072) was purchased from the China Center for Type Culture Collection. *L. par* JN-1, *L. par* E10, *L. rha*, and *E. coli* Nissle 1917 were isolated from a commercial probiotic preparation in our laboratory. *D. nitroreducens* (*D. nit*) and *E. coli* Nissle 1917 were cultured in Luria broth at 37°C in an aerobic chamber for 24–48 hours. *L. par* JN-1, *L. par* E10, and *L. rha* were cultured in De Man, Rogosa, and Sharpe medium at 37°C in an aerobic chamber for 24–48 hours.

### Cell culture and treatment

The A549 and H1975 cell lines were procured from the Stem Cell Bank (Chinese Academy of Sciences, Shanghai, China) and cultured in Roswell Park Memorial Institute 1640 (RPMI1640) (#11875093; Gibco, USA) containing 10% fetal bovine serum (FBS) (#10099141C; Gibco, USA), and 1% penicillin-streptomycin-amphotericin B solution (#C0224-100ml; Beyotime, Shanghai, China), cultured in an incubator at 37°C with 95% air and 5% $CO_2$.

The cells were seeded into 96-well plates (2,000–5,000 cells/well) and exposed to various bacteria. After incubating with bacteria at a multiplicity of infection (MOI) of 100 for 4 hours, the cells were washed twice and incubated with fresh culture medium for 48 hours. The cell counting kit-8 (#HY-K0301; MCE, USA) was used to detect the cell viability.

### Animals and treatment

Female BALB/c-nude mice aged 5 weeks ($n$ = 5 per group) were obtained from GemPharmatech Co., Ltd. (Nanjing, China). All mice were raised under specific pathogen-free conditions and had free access to diet and water. All mice were routinely checked by certified veterinarians prior to the tumor-bearing experiments. They were treated via oral gavage with *D. nitroreducens* ($1 \times 10^8$ colony-forming units/mice) or phosphate-buffered saline (PBS) daily from day 0 and subcutaneously injected with $1 \times 10^7$ A549 cancer cells on day 14. After inoculation, the tumor-bearing mice were intraperitoneally injected with 3 mg/kg oxaliplatin (#S1224; Selleck, USA) in PBS on days 10, 13, 16, 19, 22, 25, 28, 31, 34, 37, and 40. The tumor size was measured every 3 days from day 10 until the end point, and the tumor volume was calculated as length $\times$ width$^2$ $\times$ 0.5. The body weight was measured every week from week 1 until the end point. The mice were euthanized on day 40. The tumor, liver, spleen, cecum, colon, and serum were collected and prepared for subsequent analysis.

### 16S rRNA gene sequencing and data analysis

The fecal samples were used for subsequent 16S rRNA gene sequencing. The bacterial 16S rRNA gene V3–V4 region universal primer pair was used to amplify the DNA by polymerase chain reaction (61). The library and its products were sequenced using the PE300 module on the HiSeq2500 (BGI, Wuhan, China). The Divisive Amplicon Denoising Algorithm software in the Quantitative Insights into Microbial Ecology 2 package was used to assign clean tags.

Silva_species_assignment_v138 reference database was used to annotate representative ASVs clustered by tags with ≥100% similarity. Function predictions used PICRUSt2 (v2.2.0). PCoA and functional prediction were performed on the OE Cloud Platform (Shanghai OE Biotech Co., Ltd., Shanghai, China). Alpha diversity was analyzed using Chao, Ace, Shannon, Simpson, and Sobs indices. LEfSe ($P \leq 0.01$ and LDA ≥ 2) was used to analyze the differential abundance among groups at the phylum, class, order, family, genus, and species levels.

## Histological analysis

The tumor, liver, and colon were collected from the treated mice, fixed with 4% paraformaldehyde, and embedded. The tissue sections were prepared and subjected to H&E staining. For immunohistochemistry staining, the tumor sections were antigen retrieved using citrate buffer (62) and stained with Ki-67 (1:200; Servicebio, Nanjing, China) and secondary antibody (1:200; Servicebio). The sections were scanned and analyzed using Caseviewer (3DHISTECH Ltd., Budapest, Hungary).

For the quantification of the infiltration of helper T cells (CD3$^+$ CD4$^+$), macrophages (CD163$^+$ CD68$^+$), cytotoxic T cells (CD3$^+$ CD8$^+$), and Treg cells (CD4$^+$ FOXP3$^+$), the tumor sections were labeled with IF staining. The antibodies used in the study were follows: CD163 (16646-1-AP; Proteintech, Wuhan, China), CD68 (66231-2-Ig; Proteintech), CD3 (17617-1-AP; Proteintech), CD4 (ab288724; Abcam, UK), CD8 (66868-1-Ig; Proteintech), and FOXP3 (22228-1-AP; Proteintech). After dewaxing, the tissue sections were treated with the citrate antigen retrieval solution (Beyotime) and incubated with 5% bovine serum albumin solution for 30 min. The primary antibodies were added to tissue slices at 4°C overnight, followed by secondary antibodies incubated for 1 hour at room temperature. The sections were scanned and analyzed using Pannoramic MIDI (3DHISTECH Ltd.).

## Isolation of alveolar macrophages and treatment

After the mice were anesthetized and euthanized, the lungs were perfused with PBS. The lung tissues were digested with PBS containing 1 mg/mL collagenase I (Sigma-Aldrich, USA) and 50 U/mL DNase I (Sigma-Aldrich) at 37°C for 30 min. The digested issues were filtered with a 40-μm syringe filter and centrifuged at 1,500 rpm for 15 min. After red blood cell lysis, the cells were resuspended in RPMI media containing 10% FBS and seeded in 96-well plates for 1 hour before changing the medium. After 2 hours, the cells were used for assays. *D. nitroreducens* was washed with PBS and added to the cells at an MOI of 100. For the control group, PBS was added.

## Statistical analysis

All experiments were repeated at least three times, and the data were expressed as the mean ± standard deviation. Statistical analysis was performed using the Student two-tailed *t*-test (between two groups) or one-way or two-way analysis of variance (multiple comparisons). Significance was set as follows: *$P < 0.05$ and **$P < 0.01$. GraphPad Prism 9.5 (CA, USA) was used for statistical analysis.

## ACKNOWLEDGMENTS

This work received funding from the innovation team of Wuxi Health and Family Planning Commission (CXTD2021005), the Wuxi Health Commission Fund for Young Scholars (Q202269), the project of Taihu Talent Plan, and Graduate student scientific research innovation projects in Jiangsu province (KYCX22_2373).

Y. Ni, R. Li, Y. Geng, and Q. You conceived and designed the study. Y. Ni and R. Li performed the experiments. Y. Ni, X. Shen, D. Yi, Y. Ren, and F. Wang performed the data analysis and bioinformatic analysis. Y. Ni wrote the manuscript. Y. Geng revised the manuscript. Y. Geng and Q. You approved submission of the manuscript.

## AUTHOR AFFILIATIONS

[1]Department of Oncology, Affiliated Children's Hospital of Jiangnan University, Wuxi, Jiangsu, China
[2]Key Laboratory of Carbohydrate Chemistry and Biotechnology, Ministry of Education, School of Biotechnology, Jiangnan University, Wuxi, Jiangsu, China
[3]Affiliated Hospital of Jiangnan University, Wuxi, China
[4]School of Life Science and Health Engineering, Jiangnan University, Wuxi, China

## AUTHOR ORCIDs

Yan Geng   http://orcid.org/0000-0001-8861-7686
Qingjun You   http://orcid.org/0000-0003-1151-8318

## FUNDING

| Funder | Grant(s) | Author(s) |
| --- | --- | --- |
| Innovation team of Wuxi Health and Family Planning Commission | CXTD2021005 | Qingjun You |
| Wuxi Health Commission Fund for Young Scholars | Q202269 | Qingjun You |
| The project of Taihu Talent Plan | | Qingjun You |
| Graduate student scientific research innovation projects in Jiangsu province | KYCX22_2373 | Yalan Ni |

## AUTHOR CONTRIBUTIONS

Yalan Ni, Validation, Writing – original draft | Rui Li, Investigation, Validation | Xiaoyu Shen, Formal analysis | Deli Yi, Formal analysis | Yilin Ren, Formal analysis | Fudong Wang, Formal analysis | Yan Geng, Conceptualization, Writing – review and editing | Qingjun You, Conceptualization, Project administration

## DATA AVAILABILITY

All the data are available upon request. The raw data of 16S rRNA gene sequencing generated in this study have been deposited at the National Center for Biotechnology Information (BioProject ID: PRJNA1036816).

## ETHICS APPROVAL

All animal experiments were approved by the Institutional Animal Care and Use Committee of Jiangnan University (approval number no.: JN. no. 20230330b1000705[102]).

## ADDITIONAL FILES

The following material is available online.

### Supplemental Material

**Figure S1 (mSystems01323-23-s0001.tif).** H&E staining of the liver (400×).
**Figure S2 (mSystems01323-23-s0002.tif).** (A) Alpha diversity analysis including Chao, Ace, Shannon, Simpson, and Sobs indices. (B) Bar plots showing the composition and relative abundance of each sample (left) and mean relative abundance in control, Oxp, D. nit, and Oxp+ D. nit groups (right) at the family level.
**Figure S3 (mSystems01323-23-s0003.tif).** (A) Immunofluorescence staining of the helper T cell markers CD4 (pink) and CD3 (green) with DAPI nuclear staining (blue), 40×. (B) Immunofluorescence staining of cytotoxic T cell markers CD8 (pink) and CD3 (green) with DAPI nuclear staining (blue), 40×.

### Open Peer Review

**PEER REVIEW HISTORY (review-history.pdf).** An accounting of the reviewer comments and feedback.

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
