## [Reviewer comments · mSystems]

***Diaphorobacter nitroreducens* synergize with oxaliplatin to reduce tumor burden in mice with lung adenocarcinoma**

Yalan Ni, Rui Li, Xiaoyu Shen, Deli Yi, Yilin Ren, Fudong Wang, Yan Geng, and Qingjun You

Corresponding Author(s): Qingjun You, Affiliated Children's Hospital of Jiangnan University

Review Timeline:

Submission Date:	December 7, 2023
Editorial Decision:	January 30, 2024
Revision Received:	February 21, 2024
Accepted:	February 29, 2024

Editor: Michaeline Albright

Reviewer(s): The reviewers have opted to remain anonymous.

Transaction Report:

DOI: <https://doi.org/10.1128/msystems.01323-23>

Re: mSystems01323-23 (*Diaphorobacter nitroreducens* synergize with oxaliplatin to reduce tumor burden in lung adenocarcinoma mice)

Dear Prof. Qingjun You:

Thank you for the privilege of reviewing your work. Below you will find my comments, instructions from the mSystems editorial office, and the reviewer comments (attached). Overall, the scientific content is excellent, however, for publication the text needs improvement and editing with the help of someone who has professional experience with science writing in English.

Revision Guidelines

Sincerely,
Michaeline Albright
Editor
mSystems

I would like to express my gratitude to the authors for their diligent work in elucidating the roles of *Diaphorobacter nitroreducens* in synergy with oxaliplatin to reduce the tumor burden of lung adenocarcinoma. However, several areas in the article require improvement, as outlined below:

1. Some words are misspelled and need correction.
2. The phrase "...mice bearing subcutaneous A549 cells..." is incorrectly used. Instead, it is suggested to use "tumor-bearing mice."
3. The statement about alternatively activated M2 macrophages is ambiguous and needs clearer articulation.
4. Many statements lack proper references, such as the one mentioning the association between high intratumoral infiltration of CD3+ and CD8+ cells and prognosis.
5. Consider using a more concise expression, such as "CD3+CD8+," unless the authors intended to convey something different.
6. The statement about current studies focusing mainly on the preventative potential of conventional probiotics is weak; consider revising to emphasize the role of microbiota in treatment.
7. Proper credit is not given for information obtained from the literature. It is crucial to acknowledge the intellectual owners of statements. It is a grave ethical issue in the scientific community to use other information without acknowledging it.

Some of these are;

- a) *D. nitroreducens* is an aerobic, Gram-negative bacterium belonging to phylum Proteobacteria, family Comamonadaceae.
- b) Currently, an accumulating number of clinical trials using combination therapies are being conducted to find enhanced sensitization methods.
- c) Recently, it was increasingly recognized that the composition of gut microbiota is critical for the progression of cancer treatment
- d) Tumor immune cell infiltration refers to the infiltration of immune cells from blood to the tumor tissue, which can be separated from the tumor tissue.

- e) Immune cells (macrophages and T cells) may represent the primary sources of pro-inflammatory TNF α .
- f) In mice treated with probiotics, the overall structure of intestinal villi was restored, and this phenomenon was also found in our study.
- g) Additionally, gut microbiota facilitates a plethora of chemotherapy-induced immune and inflammatory responses.
- h) It has been reported that Bifidobacteria modified tumor-specific T-cell induction and increased T-cell numbers in tumor microenvironment in patients treated with immunomodulator.
8. Many words or phrases are not spelt out at first mention, pls revise.
9. You may want to consider pluralizing your figures in naming them, such as Figures 3C and D and not in form of “Figure 3C, D”.
10. Please, revise “*The gut microbial function changes always accompanied with the alternation of gut microbiota structure.*”
11. Your statement “*We furtherly double stained macrophages (CD163 and CD68), helper T cell (CD3 and CD4), cytotoxic T cell (CD3 and CD8) and Treg cell (CD4 and FOXP3) and examined cells under an immunofluorescence (IF) (Figure 5 and S3).*” Please, consider revising. It is not understood whether the authors were referring to tumor, blood, TDLN or other lymphoid organs, it is important to write clearly.
12. The statement “*D. nitroreducens was reported to be one of the top five bacterial species in the normal nonsmoker lung, but not the dominant strain in the lung of cystic fibrosis, COPD and smoker(23).*” is misleading. Are the authors referring to the lung as the locale for the microbe?
13. Pls, consider revising, the sentence “*A recent study also proposed the role of Bifidobacterium infantis in ameliorating oxaliplatin-induced toxicity by inhibiting Th1 and Th17 activation and enhancing cytotoxin regulatory T- cell activity (CD4+CD25+Fox p3+) in rats(22)*”, 2017 is not recent!
14. Regarding the concluding remark of the “Discussion” before the “Limitation”, many questions arise. The question remaining is about other immunosuppressive cells such as TAM, MDSCs. Why not assay cytokine and chemokine profiles in the TME to understand the balance between pro-and anti-

inflammatory factors, I think this would be able to explain the TME better than the suppression of Tregs alone. The complex dynamics within the TME are a bit more complex than this explaining with this scope, only macrophage and Tregs!.

15. The references need to be revised. They're not patterned/styled the same way. For example, consider pagination in the refs 6, 8, etc compared to others.

Addressing these points will enhance the clarity, accuracy, and overall quality of the article.

Editor-in-Chief

January 31, 2024

Dear Editor,

We appreciate the expert review of our manuscript (mSystems01323-23) entitled “*Diaphorobacter nitroreducens* synergize with oxaliplatin to reduce tumor burden in lung adenocarcinoma mice”. We thank the editors and reviewers for their efforts and scientific insights, and the opportunity to address the comments of the editor and reviewers. Below is a detailed point-by-point response to the editor and reviewer’s comments. We are hopeful that the reviewers and editorial board will now find the manuscript suitable for publication in *mSystems*.

Sincerely,

Qingjun You, M.D., Ph.D.

Professor

Affiliated Children's Hospital of
Jiangnan University, Department of
Oncology

299 Qingyang Road, Wuxi 214023,
P.R.China

Phone/Fax: 86-510-85918206

Email: youqingjun@jiangnan.edu.cn

Yan Geng, Ph.D.

Professor

School of Life Science and Health
Engineering

Jiangnan University

1800 Lihu Avenue, Wuxi 214122,
P.R.China

Phone/Fax: 86-510-85910116

Email : gengyan@jiangnan.edu.cn

Editor’s Comments

1. **“Some words are misspelled and need correction. The authors are advised to thoroughly read through and pay more attention.”**

Response 1: We appreciate the reviewer’s careful corrections and suggestions which have improved the paper. The misspelled words in the manuscript have been corrected.

2. **The phrase “...mice bearing subcutaneous A549 cells...” was incorrectly used. Instead, I would suggest the authors should consider “tumor-bearing mice”.**

Response 2: As the reviewer recommended, we have modified the phrase to “*tumor-bearing mice*”.

3. The statement *“Although alternatively activated M2 macrophages have traditionally been considered a poor prognosis marker (2), current studies have also shown that both M1 and M2 macrophages have good prognostic significance (3)”* is ambiguous and needs clearer articulation.

Response 3: As the reviewer recommended, we have modified the statement to *“Although M2 macrophages are traditionally thought to contribute to tumor progression (2), recent studies have also shown that the infiltration of M2 macrophages is positively associated with favorable clinical outcomes in patients with NSCLC (3).”* to indicate the important role of M2 macrophages in NSCLC.

4. Many statements lack proper referencing, such as *“Additionally, many studies had reported that the high intratumoral infiltration of CD3+ and CD8+ cells was associated with good prognosis and improved survival, while the increase of Treg cells (FOXP3+) indicated a worse prognosis (4)”*. I cannot see many studies properly referenced here!

Response 4: As the reviewer recommended, we have added appropriate references in the following statements to support them.

Statement 1: *“Additionally, many studies reported that the high intra-tumoral infiltration of cytotoxic T cells (CD3⁺ CD8⁺) was associated with good prognosis and improved survival (4, 5). In contrast, the increase in the proportion of regulatory T (Treg) cells (forkhead box protein 3-positive, FOXP3⁺) indicated a worse prognosis (6, 7).”*

4. Panahi M, Rezagholizadeh F, Mollazadehghomi S, Farhangnia P, Niya MHK, Ajdarkosh H, Tameshkel FS, Heshmati SM. 2023. The association between CD3(+) and CD8(+) tumor-infiltrating lymphocytes (TILs) and prognosis in patients with pancreatic adenocarcinoma. *Cancer Treat Res Commun* 35:100699.

5. Li F, Li C, Cai X, Xie Z, Zhou L, Cheng B, Zhong R, Xiong S, Li J, Chen Z, Yu Z, He J, Liang W. 2021. The association between CD8+ tumor-infiltrating lymphocytes and the clinical outcome of cancer immunotherapy: A systematic review and meta-analysis. *EClinicalMedicine* 41:101134.

7. Peng J, Yu Z, Xue L, Wang J, Li J, Liu D, Yang Q, Lin Y. 2018. The effect of foxp3-overexpressing Treg cells on non-small cell lung cancer cells. *Mol Med Rep* 17:5860-5868.

Statement 2: *“Many studies showed good prognostic significance for macrophages, whereas the accumulation of Treg cells in tumor suggested a worse prognosis (3, 7, 51, 52).”*

3. Rakaee M, Busund LR, Jamaly S, Paulsen EE, Richardsen E, Andersen S, Al-Saad S, Bremnes RM, Donnem T, Kilvaer TK. 2019. Prognostic Value of Macrophage Phenotypes in Resectable Non-Small Cell Lung Cancer Assessed by Multiplex Immunohistochemistry. *Neoplasia* 21:282-293.

7. Peng J, Yu Z, Xue L, Wang J, Li J, Liu D, Yang Q, Lin Y. 2018. The effect of foxp3-overexpressing Treg cells on non-small cell lung cancer cells. *Mol Med Rep* 17:5860-5868.

51. Zeng DQ, Yu YF, Ou QY, Li XY, Zhong RZ, Xie CM, Hu QG. 2016. Prognostic and predictive value of tumor-infiltrating lymphocytes for clinical therapeutic research in patients with non-small cell lung cancer. *Oncotarget* 7:13765-13781.

52. Mlika M, Saidi A, Mejri N, Abdennadher M, Haddouchi C, Labidi S, Khiari H, Boussen H,

Hsairi M, Mezni F. 2022. Prognostic impact of tumor-infiltrating lymphocytes in non-small cell lung carcinomas. *Asian Cardiovasc Thorac Ann* 30:177-184.

Statement 3: “*Currently, an increasing number of clinical trials using combination therapies are being conducted to find enhanced sensitization methods (28-30).*”.

28. Srivastava S, Furlan SN, Jaeger-Ruckstuhl CA, Sarvothama M, Berger C, Smythe KS, Garrison SM, Specht JM, Lee SM, Amezcua RA, Voillet V, Muhunthan V, Yechan-Gunja S, Pillai SPS, Rader C, Houghton AM, Pierce RH, Gottardo R, Maloney DG, Riddell SR. 2021. Immunogenic Chemotherapy Enhances Recruitment of CAR-T Cells to Lung Tumors and Improves Antitumor Efficacy when Combined with Checkpoint Blockade. *Cancer Cell* 39:193-208 e10.

29. Xin M, Lin D, Yan N, Li H, Li J, Huang Z. 2022. Oxaliplatin facilitates tumor-infiltration of T cells and natural-killer cells for enhanced tumor immunotherapy in lung cancer model. *Anticancer Drugs* 33:117-123.

30. Chen X, Lu J, Yao Y, Huang Z, Liu K, Jiang W, Li C. 2021. Effects of bevacizumab combined with oxaliplatin intrathoracic injection on tumor markers and survival rate in patients with malignant pleural effusion of lung cancer. *Am J Transl Res* 13:2899-2906.

Statement 4: “*Increased fatty acid biosynthesis could be a response to the high metabolic demand of cancer cells or an adaptation to reduced efficacy of serum-derived lipids in the tumor microenvironment (47, 48).*”

47. Jin Z, Chai YD, Hu S. 2021. Fatty Acid Metabolism and Cancer. *Adv Exp Med Biol* 1280:231-241.

48. Rohrig F, Schulze A. 2016. The multifaceted roles of fatty acid synthesis in cancer. *Nat Rev Cancer* 16:732-749.

Statement 5: “*Bifidobacterium modified tumor-specific T cell induction and increased the number of T cells in the tumor microenvironment in patients treated with immunomodulators (9).*”.

9. Sivan A, Corrales L, Hubert N, Williams JB, Aquino-Michaels K, Earley ZM, Benyamin FW, Lei YM, Jabri B, Alegre ML, Chang EB, Gajewski TF. 2015. Commensal Bifidobacterium promotes antitumor immunity and facilitates anti-PD-L1 efficacy. *Science* 350:1084-1089.

5. Consider using a more concise expression, such as "CD3+CD8+," when you mean cytotoxic T-cells, please, unless the authors intended to convey something different, which is required to be clarified here.

Response 5: As the reviewer recommended, we have modified the statements as follows.

Statement 1: “*CD3⁺CD8⁺,*” has been modified to “*cytotoxic T cells (CD3⁺ CD8⁺)*”.

Statement 2: “*macrophages (CD163 and CD68), helper T cell (CD3 and CD4), cytotoxic T cell (CD3 and CD8) and Treg cell (CD4 and FOXP3)*” has been modified to “*macrophages (CD163⁺ CD68⁺), helper T cell (CD3⁺ CD4⁺), cytotoxic T cell (CD3⁺ CD8⁺) and Treg cell (CD4⁺ FOXP3⁺)*”.

Statement 3: “*Treg cells...helper T cell and cytotoxic T cell*” has been modified to “*Treg cells (CD4⁺ FOXP3⁺) ...helper T cell (CD3⁺ CD4⁺) and cytotoxic T cell (CD3⁺ CD8⁺)*”.

Statement 4: “*helper T cell, macrophages, cytotoxic T cell and Treg cell*” has been modified to “*helper T cell (CD3⁺ CD4⁺), macrophages (CD163⁺ CD68⁺), cytotoxic T cell (CD3⁺ CD8⁺) and Treg cell (CD4⁺ FOXP3⁺)*”.

6. The statement “*However, current studies mainly focused on the preventative potential of conventional probiotics. Therefore, it is necessary to explore other potentially beneficial bacteria for synergistic treatment against cancers*” is a very weak knowledge gap; consider revising to emphasize the role of microbiota in treatment. The therapeutic/treatment involvement or consideration regarding probiotics has been repeatedly reported, the authors should delve more into the recent literature.

Response 6: As the reviewer recommended, we have modified the statement to “*Moreover, many potential next-generation probiotics are currently developed using the latest-generation sequencing and bioinformatics platforms (14). These bacteria, including Eubacterium limosum (15, 16), Enterococcus hirae (8), Enterococcus faecium (17), Collinsella aerofaciens (18, 19) and Burkholderia cepacian (20) have demonstrated promising efficacy in promoting the anticancer effects. However, further exploration and evaluation are needed to elucidated the potential role of the microbiota in effectively modulating of anticancer treatment.*”

7. Proper credit was not given for information obtained from the literature. It is crucial to acknowledge the intellectual owners of many statements in the article. It is a grave ethical issue in the scientific community to use other investigators’ information without acknowledging them.

Some of these are implicated sentences are as follow;

a) *D. nitroreducens* is an aerobic, Gram-negative bacterium belonging to phylum Proteobacteria, family Comamonadaceae.

b) Currently, an accumulating number of clinical trials using combination therapies are being conducted to find enhanced sensitization methods.

c) Recently, it was increasingly recognized that the composition of gut microbiota was critical for the progression of cancer treatment

d) Tumor immune cell infiltration refers to the infiltration of immune cells from blood to the tumor tissue, which can be separated from the tumor tissue.

e) Immune cells (macrophages and T cells) may represent the primary sources of pro-inflammatory TNF α .

f) In mice treated with probiotics, the overall structure of intestinal villi was restored, and this phenomenon was also found in our study.

g) Additionally, gut microbiota facilitates a plethora of chemotherapy-induced immune and inflammatory responses.

h) It has been reported that *Bifidobacteria* modified tumor-specific T-cell induction and increased T-cell numbers in tumor microenvironment in patients treated with immunomodulator.

Response 7: We appreciate the reviewer's expert insights and effort. As the reviewer recommended, we have added appropriate references in these statements to support them.

8. Many words or phrases are not spelt out at first mention, pls revise. Moreover, pls, put all acronyms in their correct form. The standard way of abbreviating hematoxylin and eosin stain is "H&E" and not "HE"!

Response 8: We appreciate the reviewer's suggestions. As the reviewer recommended, we have defined each abbreviation at first time mention.

9. You may want to consider pluralizing naming your figures, such as "Figures 3C and D" and not in form of "Figure 3C, D".

Response 9: As the reviewer recommended, we have modified the names of figures.

10. Please, revise the statement "*The gut microbial function changes always accompanied with the alternation of gut microbiota structure.*"

Response 10: As the reviewer recommended, we have modified the statement to "*Altered gut microbiota composition might result in metabolic dysregulation or functional changes.*"

11. In your statement "*We furtherly double stained macrophages (CD163 and CD68), helper T cell (CD3 and CD4), cytotoxic T cell (CD3 and CD8) and Treg cell (CD4 and FOXP3) and examined cells under an immunofluorescence (IF) (Figure 5 and S3).*" please, consider revising. It is not understood whether the authors were referring to tumor, blood, TDLN or other lymphoid organs, but it is important to write clearly.

Response 11: As the reviewer recommended, we have modified the statement to "*We furtherly double-stained macrophages (CD163⁺ CD68⁺), helper T cell (CD3⁺ CD4⁺), cytotoxic T cell (CD3⁺ CD8⁺) and Treg cell (CD4⁺ FOXP3⁺) of subcutaneous tumor tissues and examined the cells using IF (Figures 5 and S3).*"

12. The statement "*D. nitroreducens was reported to be one of the top five bacterial species in the normal nonsmoker lung, but not the dominant strain in the lung of cystic fibrosis, COPD and smoker (23).*" is misleading. Are the authors referring to the lung as the locale for the microbe?

Response 12: As the reviewer recommended, we have deleted the statement to avoid ambiguity.

13. Pls, consider revising, the sentence "*A recent study also proposed the role of *Bifidobacterium infantis* in ameliorating oxaliplatin-induced toxicity by*

inhibiting Th1 and Th17 activation and enhancing cytotoxin regulatory T- cell activity (CD4⁺CD25⁺Fox p3⁺) in rats (22)”, 2017 is not recent!

Response 13: As the reviewer recommended, we have modified the statement to “*Bifidobacterium infantis in ameliorating oxaliplatin-induced toxicity by inhibiting helper T cell 1 and helper T cell 17 activation and enhancing cytotoxin regulatory T cell activity (CD4⁺CD25⁺Foxp3⁺) in rats (41).*”.

14. Regarding the concluding remark of the “Discussion” before the “Limitation”, many questions arise. The question remaining is about other immunosuppressive cells such as TAM, MDSCs. Why not assay cytokine and chemokine profiles in the TME to understand the balance between pro-and anti-inflammatory factors, I think this would be able to explain the TME better than the suppression of Tregs alone. The complex dynamics within the TME are a bit more complex than this explaining with this scope, only macrophage and Tregs!

Response 14: We appreciate the reviewer’s expert insights and comments. As the reviewer recommended, we have supplemented the discussion about TAMs as follows: “*The tumor microenvironment (TME) possesses various types of immunosuppressive cells including myeloid-derived suppressor cells (MDSCs), Treg cells and tumor-associated macrophages (TAMs)(53). As one of the most abundant cell types present in TME(54), TAMs have been shown to exert important roles in the tumor therapy(55, 56). TAMs can be classified into the proinflammatory M1-type and anti-inflammatory M2-type macrophages(57). TAMs can exert a killing effect on tumor cells once activated by TNF- α and INF- γ (58). Besides, there also is wide communication between TMAs and other immune compositions, such as Treg cells, cytotoxic T cells, neutrophils, dendritic cells (DCs) and so on(59, 60).*”

15. The references need to be modified. They’re not patterned/styled the same way. For example, consider pagination in the refs 6, 8, etc compared to others.

Response 15: As the reviewer recommended, we have modified these references.

We appreciate for Reviewer’ warm work earnestly, and hope that the correction will meet with approval. Moreover, we thank International Science Editing (<http://www.internationalscienceediting.com>) for editing this manuscript. We have uploaded a "Revised Article with Changes Highlighted" file. In addition, due to changes in the author’s institute, the name of affiliated address “**Department of Oncology, Wuxi Children's Hospital, Wuxi, China**” has been replaced with “**Department of Oncology, Affiliated Children's Hospital of Jiangnan University, Wuxi, Jiangsu, China**” and the order of the first and the second affiliated address has also been adjusted. Once again, thank you very much for your comments and suggestions.

Re: mSystems01323-23R1 (*Diaphorobacter nitroreducens* synergize with oxaliplatin to reduce tumor burden in mice with lung adenocarcinoma)

Dear Prof. Qingjun You:

Your manuscript has been accepted, and I am forwarding it to the ASM production staff for publication. Your paper will first be checked to make sure all elements meet the technical requirements. ASM staff will contact you if anything needs to be revised before copyediting and production can begin. Otherwise, you will be notified when your proofs are ready to be viewed.

Cover Image Submissions: If you would like to submit a potential Cover Image, please email a file and a short legend to msystems@asmusa.org. Please note that we can only consider images that (i) the authors created or own and (ii) have not been previously published. By submitting, you agree that the image can be used under the same terms as the published article. Image File requirements: TIF/EPS, 7.5 inches wide by 8.25 inches tall (at least 2,250 pixels wide by 2,475 pixels tall), minimum 300 dpi resolution (600 dpi preferred), RGB, and no figure elements, e.g., arrows or panel labels. The legend should be a short description of the image, 1-2 sentences recommended.

Sincerely,
Michaeline Albright